# The Effects of Short-Term Visual Feedback Training on the Stability of the Roundhouse Kicking Technique in Young Karatekas

**DOI:** 10.3390/ijerph18041961

**Published:** 2021-02-18

**Authors:** Stefano Vando, Stefano Longo, Luca Cavaggioni, Lucio Maurino, Alin Larion, Pietro Luigi Invernizzi, Johnny Padulo

**Affiliations:** 1Fiamme Oro, Polizia di Stato, 00128 Rome, Italy; stefanovando@gmail.com; 2Department of Biomedical Sciences for Health (SCIBIS), Università degli Studi di Milano, 20133 Milan, Italy; stefano.longo@unimi.it (S.L.); luca.cavaggioni@unimi.it (L.C.); pietro.invernizzi1@unimi.it (P.L.I.); 3CONI-Italian Olympic Committee, Campania, 80127 Napoli, Italy; info@luciomaurino.com; 4Faculty of Physical Education and Sport, Ovidius University of Constanta, 900527 Constanta, Romania; alinlarion@yahoo.com

**Keywords:** intensive training, proprioception, postural sway, testing

## Abstract

The aim of this study was to assess the efficacy of using real-time visual feedback (VF) during a one-week balance training intervention on postural sway parameters in young karatekas. Twenty-six young male karatekas (age = 14.0 ± 2.3 years) were randomly divided into two groups: real-time VF training (VFT; *n* = 14) and control (CTRL; *n* = 12). Their center of pressure (COP) displacement (path length, COPpl; distance from origin, COPod) was assessed pre- and post-training on a Wii Balance Board platform in two positions (Flex: knee of the supporting leg slightly bent, maximum hip and leg flexion of the other leg; Kick: knee of the supporting leg slightly bent, mawashi-geri posture for the kicking leg). Both groups trained twice a day for seven days, performing a one-legged stance on the non-dominant limb in the Kick position. During the training, VFT received real-time VF of COP displacement, while CTRL looked at a fixed point. No interaction effect was found (*p* > 0.05). VFT exhibited greater changes pre- and post-training in Flex COPpl (−25.2%, *g* = 1.5), Kick COPpl (−24.1%, *g* = 1.3), and Kick COPod (−44.1%, *g* = 1.0) compared to CTRL (−0.9–−13.0%, *g*-range: 0.1–0.7). It is possible that superimposing real-time VF to a week-long balance training intervention could induce a greater sport-specific balance-training effect in young karatekas.

## 1. Introduction

Karate is a martial art where postural stability is of great importance for performance [1]. Indeed, many actions are performed on a single leg stance at maximum speed using different postures (e.g., kicking and transitions between postures). One of the main kicking techniques employed during kumite competitions (free sparring against an opponent) is a roundhouse kick named mawashi-geri [2]. Its execution requires a coordinated body segment sequence performed at the highest possible speed:i.Flexing the hip of the kicking leg with a flexed knee while standing on the supporting leg;ii.Extra-rotating the supporting leg while aligning the kicking leg with the target and stabilizing balance, slightly arching the torso;iii.Extending the knee to reach the target with the foot without a damaging impact; andiv.Flexing the knee of the kicking leg [3]. Given this complex interaction between the supporting leg, speed, trunk stability, and circular trajectory, training to achieve correct postural control is crucial for mawashi-geri efficacy.

To optimize and develop an athlete’s fundamental motor skills, static (i.e., stationary body) and dynamic (i.e., while moving) balancing ability is of great importance [4]. It has also been reported that sport disciplines requiring skilled, fast actions improve postural control [5,6,7,8]. In this context, karate has been demonstrated to represent an effective stimulus for balance control improvement [9], most likely due to a combination of practicing intense complex motor tasks and a substantial load being placed on the ankle joint. However, several studies concerning karate and postural control have been conducted in adults [9]. Therefore, there is great interest in finding training methods that aim to improve the postural stability of young karate athletes, bearing in mind that maturity and biological age may have a direct influence on balance system organization. Indeed, the somatosensory afference seems to be developed at 3–4 years of age and the visual system, as well as the vestibular component, reaches complete maturation at 15–16 years of age [10].

In sports where motor control is essential, such as karate [11,12], improvements in performance can be achieved by receiving external feedback about the movement features—so-called augmented feedback [13]. This method can be effective in both individual and team sports, and allows athletes to adjust for possible movement errors through instructions given about technical ability [13]. One particular type of augmented feedback involves vision (real-time visual feedback), which can provide this sensory information to the central nervous system, helping to reduce motor output variability [14]. Visual feedback training (VFT) has been successfully employed in different sports; for example, to help athletes improve the mechanical work against gravity in runners [15], the mechanical effectiveness of pedaling during steady-state cycling [16], or the explosive leg press maneuver [17]. VFT has been employed for improving stance stability [18]. The positive effects of VFT seem to refer to enhanced motor guidance, better focus on the task, and motivation as a result of task accomplishment (e.g., training at a higher intensity) [13]. In the context of karate, it has been shown that one session of VFT could acutely improve postural sway in young karatekas [19]. However, it is unclear whether this effect could be reflected in better performance during a more complex task, such as standing on one leg and performing a kicking action.

Recently, the Wii Balance Board (WBB, Nintendo, Kyoto, Japan) has been employed in different fields as a simple, accessible, and reliable device for assessments of bipedal balance [20,21], single-leg stance postural control [22], and muscle asymmetries [23], as well as a tool for balance training in clinical and exercise fields, with and without visual feedback [21,24,25]. The low cost and reliability of this equipment are the main advantages of its broad application in both clinical and performance-related settings.

Therefore, the aim of this study was to assess if superimposing real-time visual feedback on a one-week balance training program performed by young karatekas on the WBB could improve postural stability during a sport-specific kicking action.

## 2. Materials and Methods

### 2.1. Participants

A total of 38 young male karatekas (mean ± SD: age 13.4 ± 2.4 years; stature 156.5 ± 12.8 cm; body mass 53.0 ± 15.1 kg) participating in a one-week intensive karate training camp volunteered to take part in this study. Participants were randomly divided into two groups: real-time visual feedback training (VFT; *n* = 19, age 14.1 ± 1.8 years, stature 161.4 ± 11.2 cm, body mass 58.4 ± 11.3 kg), and a control condition, in which participants stood in front of a wall looking at a fixed point (CTRL; *n* = 19, age 13.2 ± 2.6 years, stature 154.4 ± 12.2 cm, body mass 54.1 ± 17.0 kg). Members of both groups had at least four years of karate training experience with at least two karate training sessions per week (~3 h per week), and were familiar with exercises involving a single-leg stance. During the training camp, all the athletes lived together and followed a diet provided by a sports nutritionist [26]. None of the participants underwent any specific balance training, strenuous endurance activity, or resistance training outside of their normal training program. The study conformed to the Declaration of Helsinki and subsequent updates, and it was conducted after the approval of the Ethics Committee of Ovidius University of Constanta (23/2020). The procedures, risks, and goals were explained to the participants’ tutor. In addition to receiving consent from the subjects, written parental consent was also obtained prior to subjects’ participation.

### 2.2. Experimental Setup

Participants reported to the experimental room (average temperature: 23 °C, min: 22 °C, max: 24 °C; relative humidity: 55 ± 2.3%) three times for testing procedures in the afternoon (2–4 p.m.) to avoid any circadian effects [27]. For the reliability of measurements and familiarization with procedures, all participants were assessed on a stabilometric platform on the first and second days of testing, with two days in between. The second testing day was used as a baseline measure (Pre). The stabilometric test was performed on the third testing day using the same procedure as for Pre (i.e., post-training—Post). The postural stability tests of the single-legged stance lasted 20 s on a Nintendo^TM^ WBB, with the non-dominant leg (supporting leg) and open eyes; trials were performed in random order (Latin square design) with 1 min of recovery in between [28]. The CoreMeter^TM^ software was employed to analyze the center of pressure (COP) from the point of origin of the Cartesian plane.

Two positions were assessed before and after training:
i.Knee of the supporting leg slightly bent, maximum hip and leg flexion of the other leg (Flex, Figure 1a);ii.Knee of the supporting leg slightly bent, mawashi-geri posture for the kicking leg (Kick, Figure 1b).iii.For each position, the upper limbs were positioned in guard (i.e., both closed hands close to the head). A manual goniometer was used to standardize the knee angle of the supporting leg ~155° of knee flexion (with 180° = full extension) [3].

### 2.3. Data Collection and Analysis

The WBB, validated by Clark et al. [20], contains four micro foil-type strain-gauge transducers (sampling rate = 100 Hz) located in each of the four corners of the board. The WBB was interfaced with a laptop computer via Bluetooth^®^ using custom software (CoreMeter^TM^ 0.9, Latina, Italy) and calibrated by placing a variety of known loads at different positions on the platform. Once paired successfully, the device can be accessed through the standard Bluetooth^®^ stack. The device can be interrogated at any time to read the current settings from the four strain-gauge sensors on the board, which are delivered as 16-bit integers. By taking into account the position of the sensors and the recorded values, the position of the COP can be easily calculated [26]. The WBB sensors have an internal fixed sampling rate, which we determined to be 100 Hz. Raw calibration data and raw sensor values were stored in a relational database on the local machine. This allows for flexible post-test data processing. A report-generation tool analyzed the collected data from the database and produced summary reports. The outcome measure used in this study was total COP displacement. Therefore, total COP displacement was chosen as the primary outcome measure because it is known to be a reliable and valid measure of standing balance [20].

The COP coordinates (*X*, *Y*) were calculated using the data from the four sensors on the WBB using the following equation:(1)$$\left(\begin{array}{c} X\\ Y \end{array}\right)=\;\frac{\sum_{i=1}^{4}{\mathrm{Wght}}_{{i\;}^{.}}\left(\begin{array}{c} x_{i}\\ y_{i} \end{array}\right)}{\sum_{i=1}^{4}{\mathrm{Wght}}_{i}}$$
where (*x_i_*, *y_i_*) = coordinates of each pressure sensor (*i*) in the Wii Balance Board’s reference frame; Wght_i_ = weight recorded on each sensor (*i*); and (*X*, *Y*) = coordinates of the COP [25]. After determination of the COP, its path length (COPpl) and distance from origin (COPod) were calculated automatically by the CoreMeter^TM^ software.

### 2.4. Training Protocol

The training protocol was composed of two sessions per day for seven days. The first session was performed in the morning and alternated between 1 min of balance training and 1 min of passive recovery, for a total of 5 min. The second session was held in the afternoon using the same procedure. Both groups performed a one-legged stance on the non-dominant limb while keeping the kicking leg in mawashi-geri posture on the WBB (as in Figure 1b). The VFT group could see the COP displacement in real-time and their goal was to keep it as centered as possible [19,26]. CTRL performed the same protocol as VFT without receiving any COP displacement feedback while staring at a fixed point, trying to stay as steady as possible.

### 2.5. Statistical Analysis

Data were tested for normality using the Shapiro–Wilk test. The Student’s *t*-test for independent samples was used to detect any initial differences between groups pre-test. The reliability of COPpl and COPod measurements was assessed in a randomly selected sub-sample of 15 participants by intra-class correlation coefficient (ICC) with 95% confidence interval (95% CI), and classified as follows: very high if >0.90, high if between 0.70 and 0.89, and moderate if between 0.50 and 0.69 [29]. Moreover, the standard error of measurement as percentage (SEM%) was calculated for each variable as a measure of absolute reliability [30,31]. The between-group differences in COPpl and COPod changes over time were analyzed using a two-way analysis of variance (ANOVA) with time as a repeated-measure factor (two levels: Pre- and Post-training) and group as a between-factor (two levels: VFT and CTRL). The ANOVA effect size was evaluated with partial eta squared (η_p_^2^) and classified as follows: small, <0.06; medium, 0.06–0.14; and large, >0.14 [32]. The Hedge’s *g* effect size with 95% CI was also calculated and interpreted as follows: trivial, 0.00–0.19; small, 0.20–0.59; moderate, 0.60–1.19; large, 1.20–1.99; and very large, >2.00 [33]. The level of statistical significance was set at *p* ≤ 0.05 in all comparisons. Data were analyzed using XLSTAT 12.3.01 (Addinsoft, SARL, Long Island City, NY, USA) and SPSS (IBM SPSS Statistics v. 19, Armonk, NY, USA) statistical software packages. Descriptive statistics were expressed as mean ± standard deviation (SD). Percentage differences were shown with 95% CI of the change.

## 3. Results

There were no baseline differences between the groups in age, stature, body mass, training experience, and all other variables studied (*p* > 0.05). During the experimental week, eight participants withdrew from the study (three from VFT and five from CTRL) due to personal reasons not linked to the experimental procedures. Four participants (two from VFT and two from CTRL) were deemed as outliers and removed from the study. Therefore, the sample size was reduced to *n* = 14 in VFT (age 14.4 ± 2.5 years, stature 159.2 ± 10.4 cm, body mass 54.2 ± 11.7 kg) and *n* = 12 in CTRL (age 13.5 ± 1.9 years, stature 156.9 ± 9.2 cm, body mass 52.2 ± 16.0 kg).

### 3.1. Reliability

In Flex, reliability was high for both COPpl (ICC = 0.86, 95% CI = 0.60 to 0.95; SEM% = 5.4%) and COPod (ICC = 0.96, 95% CI = 0.88 to 0.99; SEM% = 7.9%). Likewise, reliability of the Kick position was high for both COPpl (ICC = 0.91, 95% CI = 0.74 to 0.97; SEM% = 6.1%) and COPod (ICC = 0.97, 95% CI = 0.90 to 0.99; SEM% = 7.7%).

### 3.2. VFT-Induced Effects

The training-induced effects in VFT and CTRL are reported in Table 1 for both COPpl and COPod. The ANOVA did not reveal any interaction effects in the parameters analyzed (*p* range: 0.07 to 0.49, η_p_^2^ range: 0.02 to 0.15, small to medium). There was a significant effect of time in all analyzed variables (*p* range: 0.02 to <0.001, η_p_^2^ range: 0.24 to 0.56, large).

In VFT, COPpl changed by −25.2% in Flex (*g* = 1.5, 95% CI = 0.6 to 2.3, large) and by −24.1% in Kick (*g* = 1.3, 95% CI = 0.5 to 2.2, large) after intervention. COPod changed by −35.2% in Flex (*g* = 1.0, 95% CI = 0.2 to 1.8, moderate) and by −44.2% in Kick (*g* = 1.0, 95% CI = 0.3 to 1.8, moderate) compared to Pre values.

In CTRL, COPpl changed by −0.9% in Flex (*g* = 0.1, 95% CI = −0.7 to 0.9, trivial), and by −13.0% in Kick (*g* = 0.6, 95% CI = −0.2 to 1.5, moderate) after intervention. COPod changed by −11.9% in Flex (*g* = 0.7, 95% CI = −0.1 to 1.6, moderate), and by −5.6% in Kick (*g* = 0.5, 95% CI = −0.33 to 1.3, small) positions.

## 4. Discussion

This study was designed to investigate the efficacy of a one-week balance training program combined with visual feedback in improving stability during a sport-specific kicking action in young karate athletes. The main results showed that all together, COPpl and COPod changed between Pre- and Post-intervention in all tests. Despite no group × time interaction, the effect size analysis evidenced a greater impact of real-time visual feedback training compared to the control condition in COPpl and in COPod in Kick. These findings indicate that a short-time balance intervention was effective in improving specific balance in young karatekas. It is possible that this type of visual feedback could influence the magnitude of the results within such a short-time intervention.

### 4.1. Preliminary Considerations

It is worth mentioning that the WBB (combined with CoreMeter^TM^ software) could be an easy, accessible, and feasible device to assess standing balance in different environments than a laboratory setting. It could be advantageous to study different populations, as previously shown [20,26]. However, force in the horizontal axes cannot be assessed on the WBB, thus representing an inherent limitation of the device. Nonetheless, Clark et al. [20] highlighted that the force levels in the horizontal axes were quite low (rarely exceeding 5 N). Moreover, excellent concurrent validity of the WBB compared to the gold standard has been demonstrated [20,28], suggesting that, despite the inherent limitations, this device can be used effectively to assess standing balance.

COPpl and COPod provide indirect information about the balance control process and strategy. The reduction found in these values indicate an improvement in postural control during a single-leg specific-task balance test. These results suggest that young karatekas can adapt quickly to balance stimuli even within a one-week training performed twice a day, as previously demonstrated in a younger group (~10 years of age) [26].

### 4.2. Effects of VFT

The main finding of this investigation was that short-term, sport-specific balance training was able reduce COP displacement (COPpl and COPod). Despite the lack of an interaction effect, the effect size analysis evidenced a greater impact of real-time visual feedback in almost all variables. According to the present results, previous findings showed that VFT was effective in enhancing postural control [18,34,35]. Furthermore, a recent study showed significant improvements in postural sway following one session of real-time visual feedback practice in ~16 year old young karate athletes [19]. Moreover, Shin et al. [36] showed that visual feedback can be crucial in optimizing postural control in young people, which improves until 15–16 years of age due to growth and maturation per se [10].

We tested our participants in two single-leg stance positions, which are crucial for the correct execution of the mawashi-geri kick: Flex (i.e., the “loading” of the kick, with both hip and knee in a flexed position of the kicking leg) and Kick (i.e., the actual mawashi-geri posture, with the knee in an extended position). The VFT group obtained a marked decrease in both COPpl and COPod compared to CTRL. Therefore, it can be speculated that training with real-time visual feedback can be more effective than simply staring at a fixed point. We can hypothesize that in a simple task, such as being in the Flex position, the attentional demand for keeping the posture required in the CTRL group was not as high as when receiving a real-time visual feedback of COP displacement. Indeed, it has been demonstrated that the presence of an external cue (e.g., real-time COP displacement) could act as a constant reminder to keep the focus on the task during the training [37]. Interestingly, in the Kick position (i.e., complex task), both groups reduced COPpl values with a similar effect size. Since this testing position reflects the training position performed by both groups, this finding can be interpreted as training–testing specificity (i.e., equal training and testing positions).

The ANOVA did not evidence an interaction effect in any of the variables. It is likely that data variability accounted for this result. As a future perspective, in light of the different Pre–Post effect sizes, it would be interesting to assess whether, with a longer training period, the VFT and CTRL groups would exhibit similar results, or if VFT would superimpose a greater training stimulus compared to balance training alone.

For young karatekas, the improvement of basic and specific motor abilities are key features for the achievement of top fighting results [1]. Training balance with visual feedback, either real-time or at a fixed point, seems to be effective in managing better postural control during a mawashi-geri kick. Therefore, it can be suggested that incorporating these types of exercises during daily practice, particularly when the time for balance training is limited, would be beneficial. As mentioned previously, vision certainly played a great role in the training-induced adaptations seen in the present study. However, it is difficult to differentiate the effective contribution of each sensory system (visual, vestibular, somatosensory) to the final outcomes, and whether or not these systems adapted to the training stimuli. Nevertheless, the consistency between the improvements in postural sway parameters as a consequence of this short-time intervention is encouraging. Future studies may explore the effects of different balance training conditions, conducted over a long time period, and analyzing their effects on both performance parameters and on a competitive karate level.

This study has several possible limitations to be pointed out; firstly, the small sample size of participants. Nonetheless, we tried to recruit as many participants as possible during the one-week training camp, and we observed a significant reduction in the parameters measured. This point highlights that the sample size was probably sufficient for this kind of study design. Secondly, the training period was short in duration. Therefore, longer and more detailed intervention studies are needed to clarify the mechanisms responsible for the training-induced adaptations. Thirdly, the WBB is not a platform created for data collection. Nonetheless, its validity against a gold-standard reference has been demonstrated, as previously mentioned. Finally, no potential asymmetries between legs in the training-induced response were examined. It would be interesting to assess if balance training while standing on the preferred kicking leg would have led to different results.

## 5. Conclusions

Superimposing real-time visual feedback to a one-week balance training intervention improved sport-specific balance performance in young karatekas, with a greater effect size compared to balance training alone. These results highlight the potential of using the VFT method in this population of athletes.

## Figures and Tables

**Figure 1 ijerph-18-01961-f001:**
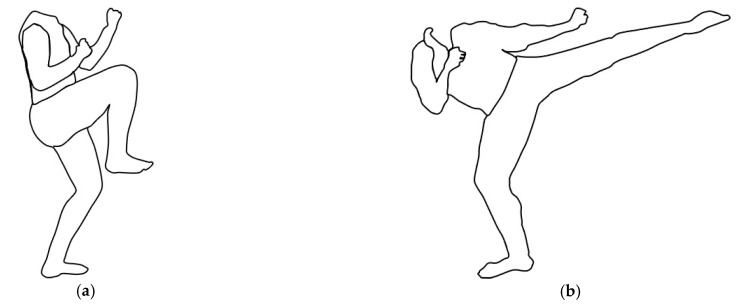
Testing position for single-leg balance. (**a**) Flex: knee of the supporting leg slightly bent, maximum hip and leg flexion of the other leg; (**b**) Kick: knee of the supporting leg slightly bent, mawashi-geri posture for the kicking leg. Note: training was performed as in (**b**).

**Table 1 ijerph-18-01961-t001:** Comparison of stabilometric data between the Pre- and Post-training in the two groups by two-way analysis of variance (ANOVA).

						ANOVA					
Condition	Group					Time			Group × Time Interaction		
		COPpl (mm)									
Flex		Pre	Post	Δ%	95% CI (%)	F	*p*	η_p_^2^	F	*p*	η_p_^2^
	VFT	166.7 ± 32.7	123.7 ± 23.3	−25.2	−31.8 to −18.6	6.22	0.02	0.24	2.69	0.12	0.12
	CTRL	176.1 ± 67.6	167.2 ± 63.8	−0.9	−22.6 to 20.8						
		COPod (cm)									
		Pre	Post	Δ%	95% CI (%)	F	*p*	η_p_^2^	F	*p*	η_p_^2^
	VFT	4.9 ± 2.5	2.8 ± 1.3	−35.2	−49.9 to −20.6	16.70	0.001	0.45	1.46	0.24	0.07
	CTRL	5.3 ± 1.8	4.1 ± 1.3	−11.9	−40.7 to 17.0						
		COPpl (mm)									
Kick		Pre	Post	Δ%	95% CI (%)	F	*p*	η_p_^2^	F	*p*	η_p_^2^
	VFT	174.8 ± 33.4	125.0 ± 26.9	−24.1	−32.9 to −15.3	25.86	<0.001	0.56	0.49	0.49	0.02
	CTRL	177.9 ± 60.5	146.0 ± 29.7	−13.0	−26.3 to 0.3						
		COPod (cm)									
		Pre	Post	Δ%	95% CI (%)	F	*p*	η_p_^2^	F	*p*	η_p_^2^
	VFT	5.2 ± 2.7	2.8 ± 1.7	−44.2	−55.8 to −32.6	18.15	<0.001	0.48	3.67	0.07	0.15
	CTRL	4.9 ± 2.3	4.0 ± 1.1	−5.6	−31.3 to 20.1						

Values are expressed as mean ± SD for the visual feedback training (VFT) and control group (CTRL). Flex: knee (~155°) of the supporting leg slightly bent, flexed hip and knee of the kicking leg; Kick: knee (~155°) of the supporting leg slightly bent, mawashi-geri posture for the kicking leg; COPpl: center of pressure path length; COPod: center of pressure distance from origin; Δ%: percentage difference between Pre- and Post-training; 95% CI (%): 95% confidence interval of the Pre–Post percentage difference; η_p_^2^: partial eta-squared.

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
