# Peer review of "The Effects of Short-Term Visual Feedback Training on the Stability of the Roundhouse Kicking Technique in Young Karatekas"

_ijerph, 2021, doi:10.3390/ijerph18041961_

Round 1

Reviewer 1 Report

Main feedback:

The abstract is quite difficult to follow and there are major revisions of grammar and construction of sentences required. The abbreviations are not clearly explained. The results are not clearly presented and the main purpose and rationale for the study cannot be determined. The introduction does not highlight the gap in the literature for this study nor how the findings can be applied to athletes. The study design is poor with only a 7 day intervention? Why so short? I am also not confident in the tool used to measure changes in COP. The Results seem to show no differences between groups but it took be a couple of reads to fully understand. The first paragraph of the Discussion is very misleading and suggests differences between the VFT and control groups? But this is not consistent with the Results section? The Discussion needs major improvements because you consistently jump back and forth between suggesting group differences and no-differences.

Some minor corrections:

Line 15: “….of a one-week balance…”

Line 17: “…participating in a…”

Line 86: 3-4 sessions?

Line 113: 155 degrees flexion?

Author Response

Reviewer #1

Authors’ response

We would like to thank the Reviewer for taking its time to critically evaluate our manuscript. We have considered every suggestion made and we have tried to accommodate the Reviewer’s comments as best as we could. Thank you for the opportunity to improve our article. We are sure that the Reviewer will consider every constructive point arising from this revision.

Main feedback:

Q1. The abstract is quite difficult to follow and there are major revisions of grammar and construction of sentences required. The abbreviations are not clearly explained. The results are not clearly presented and the main purpose and rationale for the study cannot be determined.

A1. We apologize for the lack of clarity in the Abstract. It has been completely rewritten to accommodate the Reviewer’s request and the journal guidelines (i.e., max 200 words). We hope that this version of the Abstract fully addresses the Reviewer’s criticism.

Q2. The introduction does not highlight the gap in the literature for this study nor how the findings can be applied to athletes.

A2. We agree with the Reviewer’s observation. We have tried to improve the Introduction by better describing the gap in the literature and the potential practical applications to athletes.

Q3. The study design is poor with only a 7-day intervention? Why so short?

A3. The study was realized during a summer camp, in which athletes gather and train intensively for seven days. We took the advantage of both participants availability and wanted to test the hypothesis that real-time visual feedback could help in speeding up the process of postural adaptations to balance training stimuli.

Q4. I am also not confident in the tool used to measure changes in COP.

A4. We understand the Reviewer’s concern. However, there are several reasons for the use of the WBB: the device is not expensive, it is funny, easy to use and transport. Moreover, combined with an appropriate software can provided valid data about postural sway, as already reported in previous studies (Clark et al.: 10.1016/j.gaitpost.2009.11.012; Young et al.: 10.1016/j.gaitpost.2010.10.089)

Q5. The Results seem to show no differences between groups but it took be a couple of reads to fully understand.

A5. We do apologize for the lack of clarity in reporting the results. To strengthen our results and following criticisms of another Reviewer, we revised data analysis. In detail, we excluded the outliers from both groups (two unreliable participants per group) and re-run the statistical analysis. To account for possible bias due to baseline values in balance tests, we also performed an analysis of covariance considering Pre-values as covariate. This new (and more appropriate) analysis led to different results. Now, we have demonstrated that the visual-feedback training group had significant better improvement in postural stability compared to the control group. The Statistical Analysis, Results, and Discussion sections have been modified according to the new procedures. We hope that this version of the manuscript overcomes the issues raised by the Reviewer.

Q6. The first paragraph of the Discussion is very misleading and suggests differences between the VFT and control groups? But this is not consistent with the Results section?

A6. We agree with the Reviewer’s observation. In light of the new analysis and results, the Discussion section has been modified accordingly.

Q7. The Discussion needs major improvements because you consistently jump back and forth between suggesting group differences and no-differences.

A7. We fully agree with the Reviewer’s indication. We did our best to improve the readability of the Discussion section.

Some minor corrections:

Q8. Line 15: “….of a one-week balance…”

A8. The abstract has been changed in this new version of the manuscript.

Q9. Line 17: “…participating in a…”

A9. The abstract has been changed in this new version of the manuscript.

Q10. Line 86: 3-4 sessions?

A10. Both groups had at least 4 years of karate training background with at least two karate training session per-week (total at least 3 hours per week). This has been now clarified in the text.

Q11. Line 113: 155 degrees flexion?

A11. Knee position has been better described in the text:

“~155° of knee flexion (with 180° = full extension)”

Reviewer 2 Report

Thank you to the authors for sharing their research results. The research is interesting and, what is worth emphasizing, well planned and carried out. The work is well written, it is easy to read. Requires minor adjustments:

  • Please provide the numbers in each group, gender and mean age + - SD. And include the years of experience of each group of athletes. It is recommended to make a table to include these data.
  • It would have been interesting to perform an asymmetry test in the strength levels of the lower limbs and present these data.

Author Response

Reviewer #2

Thank you to the authors for sharing their research results. The research is interesting and, what is worth emphasizing, well planned and carried out. The work is well written, it is easy to read. Requires minor adjustments

Authors’ comment

We thank the Reviewer for the kind words about the appreciation of the topic addressed in our manuscript. We are aware of the paucity of the literature on this topic especially in children, and we believe that this manuscript will contribute to fill this gap in the literature. We have considered the points raised by the Reviewer to improve the overall quality of the manuscript.

Q1. Please provide the numbers in each group, gender and mean age + - SD. And include the years of experience of each group of athletes. It is recommended to make a table to include these data.

A1. We respectfully highlight that only males were included in the study, as already mentioned in the Methods section. Moreover, following suggestions and addressing criticisms from other Reviewers, the Methods and Results sections have been modified with new parts and sentences have been added. In lights of these modifications, the groups characteristics were better described in the Results section within the text.

Q2. It would have been interesting to perform an asymmetry test in the strength levels of the lower limbs and present these data.

A2. We agree with this Reviewer’s point. Unfortunately, the training schedule during the summer camp did not allow to perform many assessments. Nonetheless, we have considered this point as valuable suggestion for future studies and included it as “limitations”.  

Reviewer 3 Report

The effects of short-term visual feedback training on stability of roundhouse kicking technique (mawashi-geri) in young karatekas

The aim of this study was to assess the efficacy of one-week balance training program on Wii Balance Board in young karatekas supported by visual feedback in improving postural stability during mawashi-geri kick technique. Participants were randomly divided in two groups: real-time visual feedback training and a control condition, in which participants stood in front of a wall looking at a fixed point. The results indicate that a short-time balance intervention was effective in improving specific balance in young karatekas, while the type of real-time visual feedback provided in the present study did not elicit greater adaptations compared to the control condition.

General comments

This study consists of a simple design to evaluate the effect of visual feedback on improving postural stability during the mawashi-geri kicking technique. There are no special design considerations, but there are some concerns that may compromise its validity. There is not enough independent variable maximization and little time to apply the treatment. And, especially, there are some important problems in relation to statistical analysis. In our opinion, the study should be repeated by modifying the shortcomings revealed in this review.

Specific comments

Lines 131-33. “total COP displacement was chosen as the primary outcome measure because it is known to be a reliable and valid measure of standing balance”

It is possible that this reliability has been provided in some study, but this does not guarantee that the measurements of the subjects of this study are also reliable. And this possibility is unfortunately confirmed later in this study.

Line 151 ff: Statistical analysis

If we have two groups and two measurements, the correct analysis would be a 2x2 repeated measures ANOVA. And, in the same way, if we only have two groups and two measures, what is the function of the Bonferroni correction?

COPpl and COPod measurements reliability was assessed by intra-class correlation coefficient with 95% confidence Interval, but the coefficient of variation is not included, and the ICC model that has been applied is not indicated. Furthermore, the results of these analyses (lines 166-7) indicate that COPpl and COPod showed moderate reliable data (ICC = 0.61, 95% CI = -0.07 - 0.86). We cannot consider these results as moderately reliable, but clearly unreliable, because of the very small value of ICC and because in the 95% interval the value zero appears, apart from the fact that the coefficient of variation, which is a very important indicator of absolute reliability, has not been analysed. These results confirm that these variables are not always reliable (lines 131-33). Consequently, changes in the dependent variables could not be attributed to the effect of the independent variable. These problems together represent a very serious drawback in considering the results of the study to be correct.

Discussion

Lines 209-11. “The main finding of this investigation was that a short-term VFT was able reduce COP displacement (COPpl and COPod) regardless of whether or not real-time visual feed-back was provided”.

These results seem to indicate that, in the end, the study does nothing to improve knowledge about the objective itself. In this sense, in addition to the issues addressed above, we believe, as we indicate as general comments, that there is not enough maximization of independent variables and little time to apply the treatment.

Line 250. “The lack of difference between groups could be due to measurement variability”

We believe that the lack of reliability, low variance maximization and short treatment time would be the most important weaknesses of the study.

Lines 265-67. “Finally, the WBB is not a platform created for data collection. Nonetheless, its validity against a gold-standard reference has been demonstrated, as previously mentioned”.

If the WBB is not a platform created for data collection, why is it used to evaluate the performance of subjects? In addition, the supposed validity of the platform is not sufficient reason to justify its use, since despite this, the measures, as we have previously indicated, may not be reliable. The instrument may be valid, but the subjects (its measures) are not, therefore, the results (measures) are no longer valid.

Author Response

Reviewer #3

The aim of this study was to assess the efficacy of one-week balance training program on Wii Balance Board in young karatekas supported by visual feedback in improving postural stability during mawashi-geri kick technique. Participants were randomly divided in two groups: real-time visual feedback training and a control condition, in which participants stood in front of a wall looking at a fixed point. The results indicate that a short-time balance intervention was effective in improving specific balance in young karatekas, while the type of real-time visual feedback provided in the present study did not elicit greater adaptations compared to the control condition.

Authors’ response.

We would like to thank the Reviewer for taking its time to critically evaluate our manuscript. We have considered every suggestion made and we have tried to accommodate the Reviewer’s comments as best as we could. Thank you for the opportunity to improve our article. We are sure that the Reviewer will consider every constructive point arising from this revision.

General comments

Q1. This study consists of a simple design to evaluate the effect of visual feedback on improving postural stability during the mawashi-geri kicking technique. There are no special design considerations, but there are some concerns that may compromise its validity. There is not enough independent variable maximization and little time to apply the treatment.

A1. We understand the points raised by the Reviewer. We did our best to set up the experiment during an intensive one-week karate summer camp, which was very congested with training sessions. Therefore, we had to limit our dependent variables to those strictly related to balance training and to use a useful, easily transportable, yet valid equipment. We are aware that the short intervention is a limitation of this study. Indeed, this has been recognized within the “study limitations” paragraph. Nonetheless, the fact that we found a training-induced decrease in postural sway during balance tests highlights that this training modality warrants further investigations with longer training periods and a more detailed description of the mechanisms potentially involved in such adaptations.

Q2. And, especially, there are some important problems in relation to statistical analysis.

A2. We thank the Reviewer to help us improve the quality of our manuscript. Regarding the statistical analysis we have tried to accommodate the Reviewer’s comments as best as we could. Please, refer to A5 below.

Q3. In our opinion, the study should be repeated by modifying the shortcomings revealed in this review.

A3. We understand that the Reviewer asked to repeat the study. However, we are afraid that this request cannot be satisfied to the COVID-19 restrictions imposed in our country and the lack of available participants. To accommodate the Reviewer’s requests, we reanalyzed both reliability and main variables. In light of the new results, the Discussion section has been modified accordingly. We hope that we fully satisfied the Reviewer’s requests.

Specific comments

Q4. Lines 131-33. “total COP displacement was chosen as the primary outcome measure because it is known to be a reliable and valid measure of standing balance”. It is possible that this reliability has been provided in some study, but this does not guarantee that the measurements of the subjects of this study are also reliable. And this possibility is unfortunately confirmed later in this study.

A4. We understand the Reviewer’s concern. As a matter of fact, another reliability study could not be done due to COVID-19 restrictions. Therefore, we have decided to review our reliability data as detailed below in Q6. The outliers were removed completely from the study. Consequently, reliability is now provided for all variables by showing ICC and 95%CI. The new results demonstrated “high” reliability with narrow (and satisfactory) 95%CI. We hope that this new analysis has provided strength to the results of this investigation.

Q5. Line 151 ff: Statistical analysis

If we have two groups and two measurements, the correct analysis would be a 2x2 repeated measures ANOVA. And, in the same way, if we only have two groups and two measures, what is the function of the Bonferroni correction?

A5. Respectfully, we partially disagree with the Reviewer’s concern about the statistical analysis. A 2x2 repeated measures ANOVA implies that both factors are “within-subjects” (i.e., repeated measures on the same group). When one factor is “between subjects” and the other factor is “within subjects”, as it occurs in the present design (two different groups), the correct model is a Mixed-model ANOVA. In our case, the “between-subjects” factor is represented by the 2 groups, whereas the “within-subjects” factor (i.e., the repeated measures) is represented by the 2 time points (in our case “time”, with two levels) factors. The “between factors” and “within factors” have been correctly identified in our model and detailed in the “Statistical Analysis” paragraph. Concerning the “Bonferroni’s correction” we agree with the Reviewer’s observation. When a 2x2 analysis is performed, the correction can be avoided. Indeed, we do apologize for wrongly reporting that Bonferroni’s correction was applied, when, in fact, it was not used. Moreover, to strengthen our results we excluded the outliers from both groups (two unreliable participants per group) and re-run the statistical analysis. To account for possible bias due to baseline values in balance tests, we also performed an analysis of covariance considering Pre-values as covariate. This new (and more appropriate) analysis led to different results. Now, we have demonstrated that the visual-feedback training group had significant better improvement in postural stability compared to the control group. The Statistical Analysis, Results, and Discussion sections have been modified according to the new procedures.

A6. COPpl and COPod measurements reliability was assessed by intra-class correlation coefficient with 95% confidence Interval, but the coefficient of variation is not included, and the ICC model that has been applied is not indicated. Furthermore, the results of these analyses (lines 166-7) indicate that COPpl and COPod showed moderate reliable data (ICC = 0.61, 95% CI = -0.07 - 0.86). We cannot consider these results as moderately reliable, but clearly unreliable, because of the very small value of ICC and because in the 95% interval the value zero appears, apart from the fact that the coefficient of variation, which is a very important indicator of absolute reliability, has not been analysed. These results confirm that these variables are not always reliable (lines 131-33). Consequently, changes in the dependent variables could not be attributed to the effect of the independent variable. These problems together represent a very serious drawback in considering the results of the study to be correct.

Q6. We agree with the Reviewer’s detailed comment. As anticipated above, we inspected the reliability data and found two unreliable participants per group. Not being able to produce additional reliability data due to COVID-19 pandemic, we have adopted this solution. Outliers were excluded from the study. Reliability analysis was re-run and the new ICCs (>0.80) with 95% were detailed for all variables in the Results section. We also calculated coefficient of variation for each variable. We hope that this new analysis, together with the ANCOVA approach on the new data fully addressed the Reviewer’s concern about the whole statistical analaysis.

Discussion

Q7. Lines 209-11. “The main finding of this investigation was that a short-term VFT was able reduce COP displacement (COPpl and COPod) regardless of whether or not real-time visual feed-back was provided”. These results seem to indicate that, in the end, the study does nothing to improve knowledge about the objective itself. In this sense, in addition to the issues addressed above, we believe, as we indicate as general comments, that there is not enough maximization of independent variables and little time to apply the treatment.

A7. We agree with the Reviewer’s comment. In light of the new results, we have modified the first paragraph of the Discussion.

Q8. Line 250. “The lack of difference between groups could be due to measurement variability”. We believe that the lack of reliability, low variance maximization and short treatment time would be the most important weaknesses of the study.

A8. As explained above, we have done our best to address reliability and statistical analysis issues correctly raised by the Reviewer’s. We have also provided justification for the few variables analyzed in the present study. We hope to have addressed the Reviewer’s concern. Furthermore, in light of the results, this part of the Discussion has been removed.

Q9. Lines 265-67. “Finally, the WBB is not a platform created for data collection. Nonetheless, its validity against a gold-standard reference has been demonstrated, as previously mentioned”. If the WBB is not a platform created for data collection, why is it used to evaluate the performance of subjects? In addition, the supposed validity of the platform is not sufficient reason to justify its use, since despite this, the measures, as we have previously indicated, may not be reliable. The instrument may be valid, but the subjects (its measures) are not, therefore, the results (measures) are no longer valid.

A9. We understand the Reviewer’s concern. There are several reasons for the use of the WBB: the device is not expensive, it is funny, easy to use and transport. Moreover, combined with an appropriate software can provided valid data about postural sway, as already reported in a previous study. As for the reliability of our sample, we have already discussed how we have done our best to overcome the correct concerns raised by the Reviewer.

Reviewer 4 Report

The paper presents an experimental study on stability during a roundhouse kick in karate after training with visual feedback.

The study was performed on 38 participants. Out of the initial group, 19 students trained with dynamic visual feedback on their center of pressure (COP), while the remaining 19 formed a control group (undergoing training but without the feedback). The results of the training were recorded using a Wii Balance Board and CoreMeter and then analyzed statistically. The authors also provided a comprehensive disscussion.

The paper is well written, features a good introduction to the topic and presents interesting results. Furthermore, the use of a widely available, inexpensive measurement equipment (Wii Balance Board), allows for reproduction and future extension of the study.

I have some minor comments. Please find the details below.

1. L69 -> change "simply" to "simple".
2. [L136]
a) add a number to the equation,
b) fix the equation -> if I understand this correctly, the fraction X / Y means X or Y (i.e.: X is computed similarly to Y). Nevertheless, the "fractional" approach is a little misleading (almost as if you compute the ratio of X by Y). I suggest to either use vectors or an entirelly different naming scheme. Also, yi on the right side is not capitalized.

Author Response

Reviewer #4

The paper presents an experimental study on stability during a roundhouse kick in karate after training with visual feedback.

The study was performed on 38 participants. Out of the initial group, 19 students trained with dynamic visual feedback on their center of pressure (COP), while the remaining 19 formed a control group (undergoing training but without the feedback). The results of the training were recorded using a Wii Balance Board and CoreMeter and then analyzed statistically. The authors also provided a comprehensive disscussion.

The paper is well written, features a good introduction to the topic and presents interesting results. Furthermore, the use of a widely available, inexpensive measurement equipment (Wii Balance Board), allows for reproduction and future extension of the study.

Authors’ response.

We would like to thank the Reviewer for taking its time to critically evaluate our manuscript. We have considered every suggestion made and we have tried to accommodate the Reviewer’s comments as best as we could. Thank you for the opportunity to improve our article. As matter of fact, following the criticisms by other Reviewers we performed new analysis that changed our conclusion. The visual-feedback training resulted in greater improvements in postural control than the control group, which led us to revise also the Discussion section. We are sure that the Reviewer will consider every constructive point arising from this revision.

I have some minor comments. Please find the details below.

Q1. L69 -> change "simply" to "simple".

A1. Done

Q2. [L136] a) add a number to the equation, b) fix the equation -> if I understand this correctly, the fraction X / Y means X or Y (i.e.: X is computed similarly to Y). Nevertheless, the "fractional" approach is a little misleading (almost as if you compute the ratio of X by Y). I suggest to either use vectors or an entirelly different naming scheme. Also, yi on the right side is not capitalized.

A2. We do apologize for the error in reporting Equation 1 according to the Young et al 2011 (https://pubmed.ncbi.nlm.nih.gov/21087865/). Now it has been fixed

Round 2

Reviewer 1 Report

The manuscript has been greatly improved. Well done on making the necessary changes to enhance the quality of the manuscript. 

Author Response

thanks for your suggestions

Reviewer 3 Report

The effects of short-term visual feedback training on stability of roundhouse kicking technique (mawashi-geri) in young karatekas

General comments

This manuscript presents practically the same problems as the first version.

One week seems a short time to test the effect of any independent variable. It is not possible to maximize the variance between the two levels of the independent variable. It is necessary to extend the experimental design time.

Neither the ICC calculation model nor the CV model are indicated. Different ICC models can show quite different ICC values.

The CV is an indicator of absolute reliability, not relative reliability. The ICC indicates relative reliability.

The lowest values of the ICC confidence interval are too low and, especially, the CVs are too high.

If we want to consider the pretest values as a covariate, we must apply a repeated measures ANOVA, which in this case would be a 2x2 ANOVA. In this way, with only one test, we can analyse the effect between and within groups, and the interaction.  In addition, it is said that in order “to take into account the possible influence of baseline values on the training-induced effects, Pre-values were inserted as covariate in the statistical model”. But then it is said that "there were no baseline differences between groups in age, stature… and all variables studied (p > 0.05)". Then, it seems that it is not necessary to apply an ANCOVA to analyse the effect of the independent variable. In any case, the assumptions of the ANCOVA model are not presented.

Author Response

Authors’ response.

We would like to thank the Reviewer for taking its time to critically evaluate our manuscript. We have considered every suggestion made and we have tried to accommodate the Reviewer’s comments as best as we could.

Q1. One week seems a short time to test the effect of any independent variable. It is not possible to maximize the variance between the two levels of the independent variable. It is necessary to extend the experimental design time.

A1. We agree with the Reviewer that one week is a short-time intervention. Nonetheless, as explained in the R1, we set up the experiment during an intensive one-week karate summer camp, which was very congested with training sessions. Therefore, we had to limit our dependent variables to those strictly related to balance training and to use a useful, easily transportable, yet valid equipment. We are aware that the short intervention is a limitation of this study. Indeed, this has been recognized within the “study limitations” paragraph. However, the fact that we found a training-induced decrease in postural sway during balance tests highlights that this training modality warrants further investigations with longer training periods and a more detailed description of the mechanisms potentially involved in such adaptations. Moreover, this research design has been used in three published papers:

  • Padulo J, Chamari K, Chaabène H, Ruscello B, Maurino L, Sylos Labini P, Migliaccio GM. The effects of one-week training camp on motor skills in Karate kids. J Sports Med Phys Fitness. 2014 Dec;54(6):715-24. Epub 2014 Oct 7.
  • Padulo J, Chaabène H, Tabben M, Haddad M, Gevat C, Vando S, Maurino L, Chaouachi A, Chamari K. The construct validity of session RPE during an intensive camp in young male Karate athletes. Muscles Ligaments Tendons J. 2014 Jul 14;4(2):121-6. eCollection 2014 Apr.
  • Vando S, Filingeri D, Maurino L, Chaabène H, Bianco A, Salernitano G, Foti C, Padulo J. Postural adaptations in preadolescent karate athletes due to a one week karate training camp. J Hum Kinet. 2013 Oct 8;38:45-52. doi: 10.2478/hukin-2013-0044. eCollection 2013.

Finally, we would like to perform additional experiments. However, the current COVID-19 pandemic does not allow to set up such study design again, nor with the same participants.

Q2. Neither the ICC calculation model nor the CV model are indicated. Different ICC models can show quite different ICC values.

A2. We do apologize for not having reported the type of ICC used. We have used two-way random effect, absolute agreement (Weir, Journal of Strength and Conditioning Research, 2005, 19(1), 231–240). This has been clarified in the text. As far as CV is concerned, and in light of Q4 (below) we have now reported the standard error of measurement as percentage (SEM%) as indicator of absolute reliability in the R2 version of the manuscript.

Q3. The CV is an indicator of absolute reliability, not relative reliability. The ICC indicates relative reliability.

A3. We do apologize for this mistake. Now, the text has been amended accordingly.

Q4. The lowest values of the ICC confidence interval are too low and, especially, the CVs are too high.

A4. We respectfully disagree with this Reviewer’s observation. The exclusion of the outliers from reliability calculation improved significantly the presented ICCs. Confidence intervals also narrowed, falling between 0.52 (the lowest value retrieved in Flex COPpl) and 0.95 (the highest value retrieved in Kick COPpl). According to Munro (2004), ICCs>0.50 can be considered as “moderate”, as also specified and referenced in the text. Therefore, no ICCs nor 95%CI were below 0.50, which can lead to state that our reliability was “high” with at least “moderate” 95%CI low boundary.

In trying to overcome the Reviewer’s concerns and considering that we cannot perform the study again, we randomly selected a sub-sample of 15 participants from our original sample without the outliers. Reliability analysis has been performed on this sub-sample. We have specified this in the Statistical Analysis section. Results have been changed according to the new analysis. SEM% values have been included. We hope that this new analysis will address the Reviewer’s concerns.

Q5. If we want to consider the pretest values as a covariate, we must apply a repeated measures ANOVA, which in this case would be a 2x2 ANOVA. In this way, with only one test, we can analyse the effect between and within groups, and the interaction. In addition, it is said that in order “to take into account the possible influence of baseline values on the training-induced effects, Pre-values were inserted as covariate in the statistical model”. But then it is said that "there were no baseline differences between groups in age, stature… and all variables studied (p > 0.05)". Then, it seems that it is not necessary to apply an ANCOVA to analyse the effect of the independent variable. In any case, the assumptions of the ANCOVA model are not presented.

A5. To accommodate the Reviewer’s request, we have re-analyzed the values by two-way ANOVA. We agree with the Reviewer that this analysis needs a repeated-measure design, and this is what we have been applying since the beginning. We clearly stated in the first version of the manuscript that there was a “within factor”, which implies a repeated-measure analysis. However, we would like to stress that we cannot write that this is an ANOVA repeated-measures analysis, because it is simply not correct. The correct definition is “mixed-model analysis of variance” due to the presence of both between- and within- factors. To stick to appropriateness of statistical analysis description and to overcome the issue raised by the Reviewer, the text has been clarified as follows:

“The differences between groups in COPpl and COPod changes over time were analyzed by a two-way analysis of variance (ANOVA) with time as repeated-measure factor (two levels: Pre- and Post-training) and group as between-factor (two levels: VFT and CTRL).”